# Pfeife: Automatic Pipeline Parallelism for PyTorch

**Ho Young Jhoo** [1]   **Chung-Kil Hur** [1]   **Nuno P. Lopes** [2][3]

## Abstract

The memory requirements of machine learning (ML) models has been growing quickly. However, the memory capacity of GPUs has not kept pace. Despite significant research on reducing the memory usage of ML models, the larger models do not fit in a single device. A popular solution to the memory capacity issue is to use multiple devices in parallel. In this paper, we focus on a particular form of parallelism called pipelining, as it offers a good balance between cost and performance for many ML models. We present Pfeife, the first tool that integrates with PyTorch to provide automatic pipelining of ML models. Pfeife intercepts the execution of models and parallelizes them transparently, requiring no manual work. We show that Pfeife can execute large models that would otherwise not run due to not fitting in a single device. Moreover, Pfeife can pipeline non-sequential models such as Stable Diffusion, which are not supported by existing pipelining parallelism tools. Pfeife outperforms state-of-the-art tools by up to 22%.

## 1. Introduction

ML models have been growing in terms of memory requirements very quickly. The memory needed just for the weight matrices has gone from 0.4 GiB in 2018 (ELMo (Peters et al., 2018)) to 754 GiB in 2024 (Llama 3.1 405B FP16 (Grattafiori et al., 2024)).

While memory requirements of ML models have increased by several orders of magnitude in just a few years, hardware has not kept up the pace. For example, NVIDIA's high-end GPUs, the most used platform for training and deploying ML models, cannot fit state-of-the-art ML models in a single device. The maximum memory capacity of the top NVIDIA GPU released this year (GB200) is only 384 GiB, still unable to fit Llama 3.1.

Several techniques have been developed to circumvent the hardware memory capacity limitations, especially using multiple devices on the same server and across servers. This encompasses multiple orthogonal techniques that can be combined, including *data parallelism*, which replicates the model across devices and splits each input batch between devices; *weight sharding*, which splits parameters across devices; and *model* and *tensor parallelism* that partition the execution of one instance of the model.

These approaches face two main challenges: scalability and usability. Data parallelism and weight sharding, for instance, eventually require full synchronization of the weights across all devices, thus require heavy communication. For model and tensor parallelism, most tools are notoriously hard to use. They often require writing models in a particular way or to annotate models in a non-trivial way. Or even to pick the partitions or identify the parallelism manually. None of these tasks are easy for humans, especially for ML practitioners, who are usually not experts in distributed systems, parallel computing, or HPC.

In this paper, we focus on pipeline parallelism, which is a particular form of model parallelism. We built Pfeife, a tool that integrates with PyTorch and that pipelines models automatically with no user intervention. PyTorch's tracing JIT compiler constructs, at run time, a data-flow graph that represents the model being executed, and it is capable of looking through a lot of Python's dynamism. This enables tools like Pfeife to grab the data-flow graph of ML models. Moreover, it allows us to intercept the execution of models and parallelize the execution in a transparent way to users.

Pfeife also allows selectively parallelizing a model in a training loop, so that it can ensemble multiple models like pre-trained frozen models. This functionality helps users to train complex models like Stable Diffusion (Rombach et al., 2022) which are traditionally not supported by automatic model parallelism tools.

The contributions of this paper are as follows:

1. A specification language for describing synchronous

---

[1]Seoul National University, Republic of Korea [2]INESC-ID / Instituto Superior Técnico - University of Lisbon, Portugal [3]FuriosaAI, Republic of Korea. Correspondence to: Ho Young Jhoo <hoyoung.jhoo@sf.snu.ac.kr>.

*Proceedings of the $42^{nd}$ International Conference on Machine Learning*, Vancouver, Canada. PMLR 267, 2025. Copyright 2025 by the author(s).

and asynchronous pipeline parallelism that can simulate broad types of pipeline schedules.

2. An efficient algorithm that finds good pipeline schedules automatically from data-flow graphs with thousands of operations.

3. Investigation into the optimal workload distribution conditions. We show that unbalanced schedules with prefetching are often the optimal solution, contrary to the popular belief that work must be distributed evenly across devices.

4. Pfeife: our tool that integrates with PyTorch to provide an automated and transparent way to pipeline models across multiple devices. Pfeife offers three key functionalities: (1) fine-grained, operation-level pipeline parallelism, (2) an implementation of the scheduling algorithm that generates schedules in our language, and (3) a runtime to execute the schedules with partial pipeline parallelism and data parallelism.

We show that Pfeife is capable of running large ML models that do not fit in a single device due to memory capacity constraints. Moreover, we show that Pfeife outperforms state-of-the-art pipelining tools by up to 21% on large models, while being fully automatic.

## 2. Background and Related Work

There are many techniques to parallelize large ML models. We give a brief overview, with a special focus on pipelining.

### 2.1. Data Parallelism and Weight Sharding

*Data parallelism* (DP) replicates the model and splits an input batch into multiple mini-batches. Each device then processes a different slice of the input. Since only the size of the activations depends on the size of the mini-batch, DP reduces the size of the activations but not of the weights.

Sometimes a single weight does not fit in a single device/core's memory. For that case, weight sharding spreads the weights across devices and gathers them on demand (Rajbhandari et al., 2020; Wang et al., 2023b; Xu et al., 2021b; Zhao et al., 2023). Sharding reduces the number of duplicate copies of weights and enables sharing parts of weights across parallel instances, but it incurs in additional communication.

Data parallelism is the simplest and easiest way to parallelize a model, but it requires full synchronization (i.e., all-reduce) of gradients between devices since it replicates the weights. Its communication workload is linear in the size of weights, and thus requires a high-bandwidth network between devices.

### 2.2. Model and Tensor Parallelism

Any partition of a model to run whole operations in separate devices is called *model parallelism*.

*Tensor parallelism* (TP) splits individual operations and distributes the operations with sliced input or weights to multiple devices (Bian et al., 2021; Xu et al., 2021a; Wang et al., 2023a; Sousa et al., 2023; Singh et al., 2023). It can distribute any kind of data but requires per-operation gather/scatter communication and it is hard to automate.

Mixture-of-experts (MoE) selectively activates a subset of layers per input (Shazeer et al., 2017; Rajbhandari et al., 2022; Masoudnia & Ebrahimpour, 2014), allowing experts to be placed on separate devices and run in parallel. Large-scale frameworks increasingly combine such techniques to scale to thousands of GPUs (Jiang et al., 2024).

### 2.3. Pipeline Parallelism

Pipeline parallelism (this paper) is a particular form of model parallelism. It slices the data-flow graph of a model and executes each sub-graph in a different device.

Pipeline parallelism is an effective technique to run large models by distributing the weights and activations across devices. However, it suffers from having devices idle at times, the so-called *pipeline bubbles*.

After PipeDream (Narayanan et al., 2019) and GPipe (Huang et al., 2019) introduced the concept of pipeline parallelism for ML models, a number of approaches have appeared to overcome some of the drawbacks of pipeline parallelism, including more effective communication/computation overlap (Chen et al., 2024a).

DAPPLE (Fan et al., 2021) uses 1F1B schedules to reduce peak memory, despite noting that uneven schedules can yield better performance. AutoPipe (Liu et al., 2022) applies critical path analysis like Pfeife, but only for 1F1B without prefetching and assumes balanced partitioning is optimal. It also partitions at sub-layer granularity, whereas we support per-operation slicing.

There are other schedules that reduce pipeline bubbles. For example, the looped pipeline from TeraPipe (Li et al., 2021), BFS (Lamy-Poirier, 2023), and AIAO (Li et al., 2023a) all reduce the initial bubble and can hide gradient synchronization latency in data parallelism. The kFkB schedule (Wang et al., 2023c) aims at hiding communication latency with a cluster of $k$ forward passes, and Chimera (Li & Hoefler, 2021) uses bi-directional pipelines to increase utilization.

In terms of implementations, PiPPy (Reed et al., 2022) uses the TorchFX tracer to generate a graph, but users must manually specify cutpoints. Slapo (Chen et al., 2024b) provides a more flexible scheduling language. Varuna (Athlur et al.,

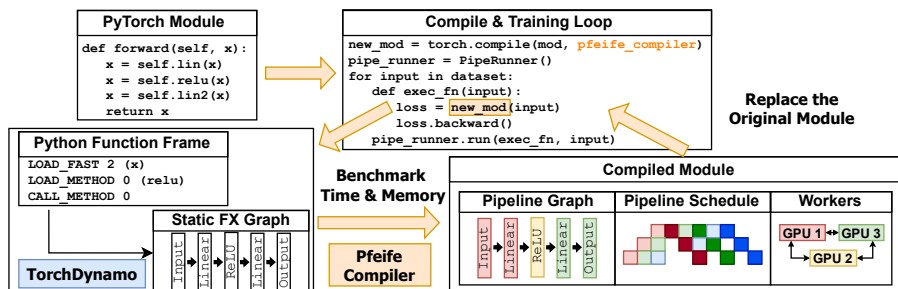

Figure 1: Overview of the architecture of Pfeife.

2022) runs on vanilla PyTorch but requires users to manually inject no-op cutpoint layers.

**Hybrid Parallelism** It is possible to combine multiple kinds of parallelism. 3D parallelism combines DP, TP, and PP for the same model.

Megatron-LM (Narayanan et al., 2021) trained transformer-like models using 3D-parallelism with the looped 1F1B pipeline. DeepSpeed (Rajbhandari et al., 2020) supports 3D-parallelism with ZeRO-DP. However, their support for TP and PP requires manual work.

Colossal-AI (Li et al., 2023b) and Merak (Lai et al., 2023) support automatic 3D-parallelism, but they are limited to a few selected transformer models of HuggingFace (Wolf et al., 2020).

Alpa (Zheng et al., 2022) and Sagemaker (Karakus et al., 2021) use several search strategies to optimize the performance of 3D-parallelism, but users have to rewrite the model using their libraries, or compile the PyTorch module with TorchScript (DeVito, 2022).

Recently, PyTorch revealed TorchTitan (Liang et al., 2024), which enables 3D parallelism in PyTorch, but is limited to train LLMs, especially Transformer models and requires manual parallelization.

There are also other parallelism tools (Jia et al., 2019; Xu et al., 2021b) that do not perfectly fit into this classification.

# 3. Pfeife

## 3.1. Architecture Overview

Figure 1 shows the architecture of Pfeife. ML models are written in plain PyTorch. They are then compiled using PyTorch 2's `torch.compile` (Ansel et al., 2024), as it is now common.

TorchDynamo is PyTorch 2's new tracing JIT compiler in `torch.compile`. TorchDynamo captures a static data-flow graph (DFG) of a PyTorch model from its function frame through symbolic execution at run time. It also allows us to intercept the execution of models and replace the original model with a parallel version of them.

Pfeife is implemented as a backend for `torch.compile`. After getting the DFG, Pfeife distributes the operations in the graph through the available devices. It then builds a pipeline schedule so that it maximizes throughput while still fitting in the memory capacity of each device.

Once a schedule is generated, Pfeife replaces the original model. Since Pfeife captures and parallelizes the actual execution of PyTorch APIs, the tool works with any user-defined model. Execution is parallelized transparently to the user, except that it will execute a large model that would otherwise go out of memory.

The main training loop is expressed as a single closure `exec_fn`. From the closure, the user can mix a pipeline module with any kind of model, such as pretrained frozen models before or after the main module. `PipeRunner` will call the closure with a set of micro-batches, catch the input entered into the pipelined model `new_mod`, and execute a pipeline schedule through the workers. This design allows pipeline parallelism for general models and use cases, whereas existing frameworks usually require a fully linearized forward pass from input to loss calculation. The appendix contains more details about the implementation.

## 3.2. Problem Setting

**Graph Construction** A core part of pipeline parallelism is how we distribute the set of operations of a model through multiple devices. Since a device can only execute one function at a time, we must know the order in which the functions have to be run on each device, and what results have to be sent to the following device. That order and the data dependencies can be expressed as a graph, which we call a *computation graph*. A computation graph is expressed as a TorchFX graph generated by TorchDynamo.

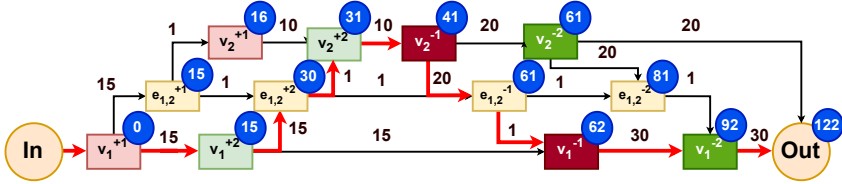

Figure 2: Example dependency graph with two mini-batches and two nodes. The computation time of $v_1$ and $v_2$ is, respectively, 15 and 10, communication time is 1, and backward nodes take twice the forward time. The red line is the critical path, and blue circles indicate the longest distance from the source node (In).

Computation graphs of large ML models can have thousands of nodes. To reduce the search space, we group several vertices together and treat them as a single node in a reduced graph, referred to as the *fused graph* $\mathcal{G} = (\mathcal{V}, \mathcal{E})$. Producing a good fused graph is key to balance scheduling time and model running time. We call this process slicing a graph, and we delve into this in Section 3.4.

**Scheduling of Pipeline Parallelism** Pipeline parallelism involves executing multiple forward and backward passes across devices according to a predefined *schedule*. We assume that each device can only run a single function (i.e., a vertex in a fused graph) on a single mini-batch at a time. We now formalize the notions of a schedule and the associated operations.

**Definition 3.1** (Schedule). A *Schedule* $S = (P, R)$ defines the order of execution of a fused graph $\mathcal{G} = (\mathcal{V}, \mathcal{E})$. Let device operations be $P : \mathcal{D} \to \mathcal{P}(Op)$, where $\mathcal{D}$ is a set of devices and $\mathcal{P}(Op)$ is the set of operations to run on each device. Operation dependency $R = \mathcal{P}(Op \times Op)$ is the set of dependencies (partial order) between operations.

$P$ denotes the set of computation and communication operations assigned to each device and mini-batch. $R$ represents the set of dependencies between these operations. These dependencies are either automatically derived from the structure of the fused graph, or manually specified by the user for, e.g., synchronization or reducing memory usage.

The set of operations, $Op$, consists of two primitive types: (1) forward or backward computation of the $i$-th vertex $v_i$ with the $n$-th mini-batch, denoted as $v_i^{\pm n} = (v_i, n, \texttt{F/B})$, where $v_i^{+n}$ represents forward computation and $v_i^{-n}$ represents backward computation; and (2) communication between two nodes $v_i^{\pm n}$ and $v_j^{\pm n}$ connected by an edge $e_{i,j}^{\pm n}$ for the $n$-th mini-batch, denoted as $e_{i,j}^{\pm n} = (e_{i,j}, n, \texttt{F/B})$.

By combining a fused graph and a schedule, we can simulate most scheduling strategies for pipeline parallelism described in the literature. We give some examples in Section 3.4.

**Operation Dependency Graph** Since $R$ is a partial order of operations in a schedule, we can derive a dependency graph of all the $Op$s by interpreting the orderings as edges. Figure 2 shows an example of that graph, which we refer to as the *operation dependency graph*, or *dependency graph*.

For simplicity, we add dependencies based on the mini-batch index without loss of generality. Also, we assume that communication of output values is executed right after the computation is finished. Existence of a dependency between two communication nodes $(e_1, e_2)$ is governed by the number of the communication channel. We assume that there is a single channel (e.g., PCIe or NVLink) between two devices. The order between $e_1$ and $e_2$ is decided by the scheduling algorithm, given the network topology.

Some implementations of pipeline parallelism such as torchgpipe (Kim et al., 2020) strictly divide the computation and communication phases. By assigning more dependencies, we can simulate synchronization operations between two devices. Considering the graph of Figure 2, if we explicitly set cross dependencies like $e_{1,2}^{+1} \to v_1^{+2}$ and $v_2^{+1} \to e_{1,2}^{+2}$, then these work like a synchronization barrier which forces communication to be executed only after all the computation is finished.

### 3.3. Cost Model

To achieve our goal of reducing the total execution time of models, we need a cost model to estimate the running time and the peak memory usage of a given graph slicing and schedule, and use it as the objective for our optimization algorithm.

Following the strategy of DAPPLE (Fan et al., 2021), we use a profiler to measure the computation time and memory consumption of each operation in a computation graph before pipelining the model. We then use a planner to search for the optimal graph slicing and execution order using the results of the profiler.

Unlike previous pipeline planners, our cost model is based on critical path analysis and thus is not dependent on any specific pipeline schedule nor needs an in-depth analysis of the schedule to build a heuristic. It is also agnostic to the structure of the model or its graph. Because of this generality, our planner found several interesting pipelines

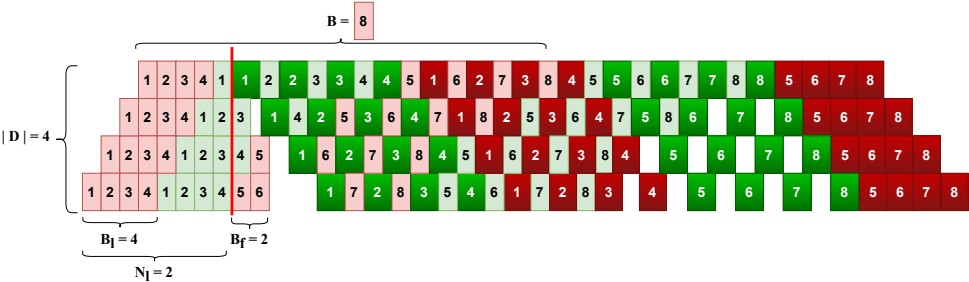

Figure 3: A looped 1F1B schedule with parameters $|\mathcal{D}| = 4$, $B = 8$, $B_l = 4$, $\vec{B_f} = (2, 2, 1, 0)$, $N_l = 2$. Each row represents a device. The x-axis is time. Light-colored (resp. shaded) boxes represent the forward (resp. backward) pass. Red/green colors indicate each of the halves of each pass.

schedules that are faster than state-of-the-art frameworks.

For the communication time, we run a set of micro-benchmarks offline to measure throughput and latency of sending data between all pairs of devices. This information is then used by the planner to estimate the communication time based on source/destination devices and data size.

**Running Time Estimation**   We can interpret dependency graphs as flow graphs with a source and a sink nodes. We want to know the earliest possible starting time of the sink node. To achieve this, we have to compute the earliest possible starting time of each of the individual nodes.

Nodes can have multiple parents. For example, in Figure 2, $v_2^{+2}$ has two parents: $v_2^{+1}$ and $e_{1,2}^{+2}$. The child node's possible starting time (31) is the maximum of the parent's ending time (start + execution times). This can be viewed as a vertex weight (starting time, blue circles) + edge weight (execution time). The running time of operations is obtained by running each operation at a time on the target device and measuring the time it takes.

Since the dependency graph is a DAG, we can use, e.g., Dijkstra's shortest path algorithm with negated weights to find the longest path, which corresponds to the estimated run time of the model. By finding the longest path from the source to the sink node, we can find the critical path and the estimated running time of a single iteration of the schedule.

**Linearization and Graph Slicing by Split Points**   Slicing a computation graph is one of the most important tasks when generating a parallel schedule. The issue is that slicing an arbitrary computation graph is equivalent to the min-cut problem, and thus it is NP-hard. However, we can simplify the decision of where to split the model by *linearizing* the graph and finding a set of *split points* in the linearized graph.

Linearizing a graph is equal to selecting one of the partial orders of the computation graph. It can be easily found through a topological sort since the computation graph is a DAG.

Selecting a particular topological order when the computation graph contains parallel branches is obviously an approximation and may not yield the optimal running time. However, most current models, such as Transformer-based models or CNNs, have a strict total order between their operations or only a few negligible parallel operations. Therefore, we opted to use the order of operations generated by the TorchDynamo compiler. We show later that this approximation yields good results in practice.

The remaining job consists in slicing the linearized sequence of nodes, i.e., selecting split points. The detailed algorithm is described in the next section.

### 3.4. Pipeline Parameters and Optimization Algorithms

We now describe how Pfeife generates fused graphs and schedules.

**Looped Training Schedule**   Any ML model has many valid pipelining schedules. Trying to find the optimal schedule at run time without any constraint is not viable as current solving techniques do not scale. Inspired by previous work, we reduce the search space to finding the following parameters for the looped 1F1B schedule (Narayanan et al., 2021):

- ($B$) Total batch count: Number of mini-batches

- ($N_l$) Loop count: How many times the forward loop is executed.

- ($B_l$) Loop batch count: How many mini-batches go through the forward pass of a single stage.

- ($\vec{B_f}$) Prefetch batch count: A list with the number of forward passes each device runs in addition to $B_l$ before it runs its first backward pass. $|\vec{B_f}| = |\mathcal{D}|$.

Obviously, each set of parameters offers a different tradeoff in terms of performance and memory consumption. With

this setting, we can express the schedules proposed in previous work including BFS (Lamy-Poirier, 2023) and looped-1F1B (Narayanan et al., 2021). See Appendix B) for the tradeoffs and examples.

Figure 3 shows an example schedule with 8 mini-batches and 4 devices. We sliced the graph into $N_l \cdot |\mathcal{D}|$ nodes and let the forward pass *loop* the sequence of devices. This makes the length of a single forward pass equal to $1/N_l$ stages, so it reduces the pipeline bubble in the beginning (i.e., the top-left idle zone in the figure).

One of our major findings is discovering the role of prefetching forward passes before the first backward pass specified as $\vec{B_f}$, as we explain next.

**Co-optimization of Slicing and Scheduling**  Both graph slicing and scheduling matter to reduce latency and peak memory consumption. However, existing planners choose only one part as their optimization target. They usually fix a schedule and then find the optimal graph for that, or equally slice the graph first and then find the best schedule for that slicing.

Our algorithm optimizes both slicing and scheduling at the same time so that it can find the best pair of scheduling and slicing. We use a superset of the schedules proposed in previous work and find the best graph slicing for each schedule.

Algorithm 1 shows the pseudo-code. We first generate a set of schedules and then find split points for each schedule based on our cost model (line 4). The set of schedules is given as a set of scheduling parameters $P_{set} = (B, N_l, B_l, \vec{B_f})$.

We find the split points for each schedule using beam search. The initial set of split points $\vec{p}$ has the sum of the weights and activations evenly distributed across devices (line 5). Then, we make a beam set $K$ that has a maximum size $L_k$, and insert a pair of latency and split points $(T, \vec{p})$ calculated from the computation graph. Next, we try to improve each set of split points using the following steps:

1. Take out a set of split points in the beam set (line 9).

2. Move one of the points by 1 to 10 nodes (line 10).

3. For each point from (2), move the other points proportional to the movement of $(p_s, k)$ (lines 11-12).

4. Calculate the latency and memory consumption for the split points from (2) and (3) (line 14).

5. If the latency is improved and the peak memory does not exceed the memory capacity, insert the new split points in the beam set (lines 15-17).

6. Select the best $L_K$ valid split points in the beam set (lines 18-21).

---

**Algorithm 1** Graph-schedule co-optimization.

1: **Input:** Computation graph $G$, batch count $B$.
2: **Output:** Fused graph $\mathcal{G}_{min}$, dependency graph $S_{min}$, and pipeline latency $T_{min}$.
3: $T_k, T_{min} \leftarrow \infty, \infty$
4: **for** $P_{set} \leftarrow possible\_param(|\mathcal{D}|, B)$ **do**
5:      $\vec{p} \leftarrow (p_1, \cdots, p_n) = init\_split\_pos(G)$
6:      $T, M_p \leftarrow calc\_time\_mem(G, P_{set}, \vec{p})$
7:      $K \leftarrow \{(T, \vec{p})\}$
8:      **while** $K$ is changing **do**
9:          **for** $(\_, (p_1, \cdots, p_n)) \in K$ **do**
10:            **for** $1 \leq s \leq n, 0 < |k| \leq 10$ **do**
11:              $\vec{p}_{single} \leftarrow (p_1, \cdots, p_s + k, \cdots, p_n)$
12:              $\vec{p}_{prop} \leftarrow (p_1 + k_1, \cdots, p_s + k, \cdots, p_n + k_n)$
13:              **for** $\vec{p} \leftarrow \{\vec{p}_{single}, \vec{p}_{prop}\}$ **do**
14:                 $T', M_p \leftarrow calc\_time\_mem(G, P_{set}, \vec{p})$
15:                 **if** $T' < T_k$ **and** $M_p < max\_mem(\mathcal{D})$ **then**
16:                    $K \leftarrow K \cup \{(T', \vec{p})\}$
17:                 **end if**
18:                 **if** $|K| > L_K$ **then**
19:                    $K \leftarrow select\_K\_min(K, L_K)$
20:                    $T_k \leftarrow max(\{T \mid (T, \_) \in K\})$
21:                 **end if**
22:              **end for**
23:            **end for**
24:          **end for**
25:      **end while**
26:      $T', \vec{p}_{min} \leftarrow min(\{(T, \vec{p}) \in K\})$
27:      **if** $T' < T_{min}$ **then**
28:          $T_{min} \leftarrow T'$
29:          $\mathcal{G}_{min}, S_{min} \leftarrow build\_graphs(G, P_{set}, \vec{p}_{min})$
30:      **end if**
31: **end for**

---

7. Repeat (2)-(6) until the set of split points does not change (line 25).

The number of movements in (3) is $round\left(\frac{(p_i - p_{i-1})(p_s + k)}{p_s}\right)$ if $i < s$ and $round\left(\frac{(p_{i+1} - p_i)(|V(G)| - p_s)}{|V(G)| - p_s - k}\right)$ otherwise. For all pairs of schedule and split points, we select the pair that has the lowest running time (lines 26-30).

The purpose of steps (2) and (3) is to fill the pipeline bubble below the schedule by assigning more operations at the beginning of the pipeline and fewer at the last devices.

For example, Figure 4 shows the overall steps with the actual split points and structure. From the first to the last schedule, the beam searcher moved the first and then the second points and finally found the optimal split points (6, 11).

The running time can be further improved with prefetching as shown in the last schedule. If we focus in the middle of the schedule, we see that it is $v_2^{+3}$ that is increasing the

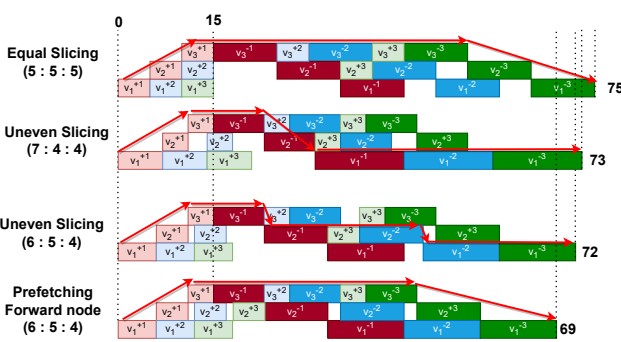

Figure 4: Latency comparison of four 1F1B schedules with 15 ops and 3 devices. The number of operations for each device is shown on the left for each schedule; the latency is shown on the right. Critical paths are shown in red.

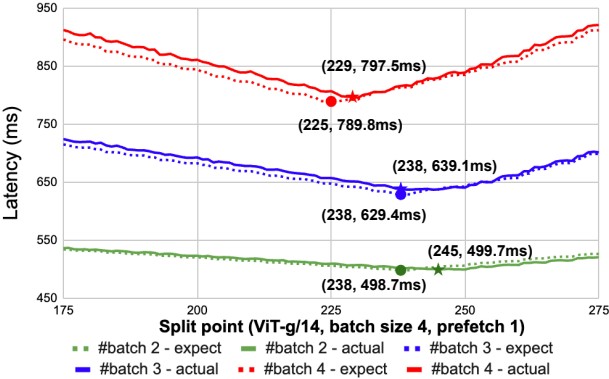

Figure 5: Expected vs actual latency of the ViT-g/14 model (407 nodes) with 2 devices. Bottom to top: 2-4 mini-batches. Circles and stars indicate the minimum expected and actual running time, respectively.

critical path. Moving it to earlier reduces the critical path.

This schedule shows that uneven slicing changes the critical path and reduces the length of the pipeline bubbles between the forward and backward passes. This is in contrast with the popular belief that even slices are optimal.

In our benchmark suite, beam search with 4 batches, 11 split points, and 406 vertices (ViT-g/14 with $N_l = 3$) completes within 10 seconds. For the largest model (StableDiffusion-XL, 1645 vertices, 8 devices), with $N_l = 2$ (15 split points) it takes less than 5 minutes, and with $N_l = 3$ (23 split points) it takes about 10 minutes. In most cases, the whole optimization process finishes within a few seconds.

## 4. Evaluation

We evaluate Pfeife[1] in three ways: (1) applicability of the approach, (2) accuracy of cost estimations, and (3) end-to-end performance comparison with existing frameworks.

We used two servers for the experiments. For coverage and correctness, we used a small server with 8x NVIDIA RTX 3090 24 GiB GPUs with 4 NVLink connections. For the end-to-end experiments, we used a larger server with 8x A100 40GB GPUs with NVSwitch. The main reason for splitting the work across two machines was because we did not have sufficient compute time available for validation in the A100 cluster.

### 4.1. ML Model Coverage

To check how applicable Pfeife is, we used Torch-Bench (Hao et al., 2023), which is the official PyTorch benchmark suite. It includes a wide range of models.

We selected 49 trainable models that allow customization of the mini-batch size and that can be run both on CPUs

and GPUs, so we have a baseline to compare against. We excluded models that do not support training by default, and models with quantized integer weights or that are too small.

We run the models with and without Pfeife and compared the results. As expected, 37 out of 49 models run successfully and produce the exact same output with and without pipelining, including complex models such as `pytorch_unet`, `hf_clip`, and `hf_Bert_large`.

Of the remaining models, 11 cannot be compiled with `torch.compile` due to issues within PyTorch itself, and one model (`timm_efficientnet`) produces incorrect results. Appendix C provides further details. These results show that Pfeife can successfully run a broad range of state-of-the-art ML models.

### 4.2. Cost Estimations and Effect of Scheduling Parameters and Split Points

We now study how the scheduling parameters affect the performance and peak memory consumption. See Appendix E for more details.

Table 5 summarizes the results. We used two models: Vision Transformer (ViT-g/14) (Zhai et al., 2022) and GPT2-large (Radford et al., 2019) with 512 tokens per batch. We observe that Pfeife safely overestimates the peak memory usage for every device (to avoid unexpected OOMs). Also, the estimated value is at most 10% higher than the actual.

Figure 5 presents a study on how the choice of split points affects the running time. Two observations are noteworthy: (1) the predicted and optimal split points yield similar execution times, demonstrating the effectiveness of Pfeife's heuristics, and (2) the optimal split is not at the midpoint. The model has 407 nodes, while the best split occurs between nodes 229–245, indicating that the first device handles

---

[1]Available at https://github.com/MerHS/pfeife.

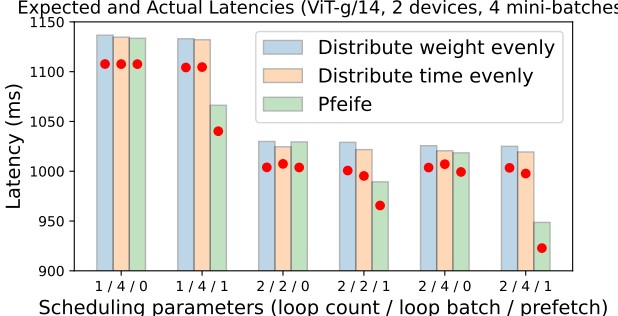

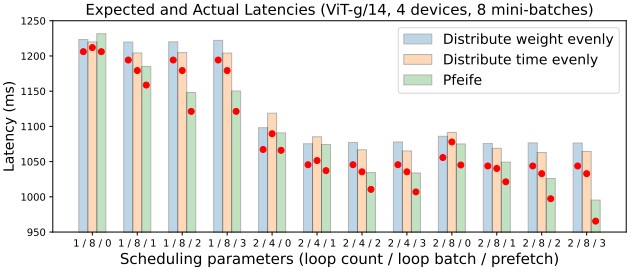

Figure 6: Expected (red points) vs actual (bars) latency of ViT-g/14 with 2 and 4 devices by changing the scheduling parameters and split points.

a larger portion of the computation.

Figure 6 shows the per-device running time of schedules with varying loop count ($L_n$), loop batch count ($B_l$), prefetch batch count ($B_f$), and split points. For all scheduling parameters, the actual latency follows the expected latency with a small (30-40ms) difference at most (I/O overhead). Therefore, Pfeife can select the optimal scheduling parameters by simply choosing the parameters which show the minimal expected latency.

We also compare three slicing strategies: distribute weights uniformly (DeepSpeed), distribute operations by running time uniformly (Alpa), and Pfeife's uneven slicing. For all cases with prefetching, Pfeife's uneven slicing outperforms the other strategies. We find that prefetching is necessary to take advantage of uneven slicing.

### 4.3. End-to-End Comparison

For the end-to-end experiments, we compared Pfeife with two semi-automatic PyTorch parallelization frameworks: DeepSpeed (Rajbhandari et al., 2020), and Colossal-AI (Li et al., 2023b). We used 3 HuggingFace models for benchmarking, namely a vision model (ViT-g/14) (Zhai et al., 2022), an LLM (Llama2-7B) (Touvron et al., 2023), and a diffusion model (Stable Diffusion-XL) (Podell et al., 2023) as an example of a non-sequential model. Note that the weights of Llama and SDXL do not fit in a single GPU.

Table 1: Throughput comparison of pipeline parallelism (item/s). $|\mathcal{D}|$: number of devices. $B$: number of micro-batches. C-AI: Colossal-AI. Higher is better. (* ZeRO 2 & 3)

| Model | $|\mathcal{D}|, B$ | DeepSpeed | C-AI | Pfeife |
|---|---|---|---|---|
| ViT | 2, 2 | 72.0 | 70.9 | **84.8** (+18%) |
| ViT | 2, 4 | 86.5 | 86.1 | **92.8** (+7.3%) |
| ViT | 4, 4 | 124.5 | 122.6 | **151.4** (+22%) |
| ViT | 4, 8 | 157.7 | 139.7 | **161.5** (+2.4%) |
| ViT | 8, 8 | 230.9 | 197.0 | **265.7** (+15%) |
| ViT | 8, 16 | **297.7** | 216.7 | 282.2 (-5.5%) |
| LLama | 8, 8 | OOM(*) | 6.80 | **7.46** (+10%) |
| LLama | 8, 16 | OOM(*) | 8.69 | **8.81** (+1.4%) |

Table 2: Throughput comparison of SDXL (item/s). Batch count is equal to the number of devices (except FP16 x2, which uses twice more batches). DP, PP: number of devices in the groups of DP and PP.

| Method | DP | PP | FP16 | FP16 x2 | FP32 |
|---|---|---|---|---|---|
| ZeRO2 | 2 | 1 | 3.78 | OOM | OOM |
| ZeRO3 | 2 | 1 | 0.81 | 0.94 | OOM |
| Pfeife | 1 | 2 | 3.72 | 4.74 | 2.38 |
| ZeRO2 | 4 | 1 | 9.12 | 10.8 | 5.32 |
| ZeRO3 | 4 | 1 | 2.01 | 1.84 | 2.01 |
| Pfeife | 1 | 4 | 7.12 | 9.32 | 3.98 |
| Pfeife | 2 | 2 | 7.06 | 8.67 | 4.54 |
| ZeRO2 | 8 | 1 | 19.8 | 19.2 | 11.2 |
| ZeRO3 | 8 | 1 | 3.82 | 3.35 | 3.22 |
| Pfeife | 1 | 8 | 12.1 | 15.4 | 6.09 |
| Pfeife | 2 | 4 | 13.9 | 15.9 | 7.32 |
| Pfeife | 4 | 2 | 13.8 | 17.2 | 9.02 |

None of the state-of-the-art tools offers fully automatic pipeline parallelism, unlike Pfeife. Colossal-AI supports just a few selected models and DeepSpeed requires a handwritten, linearized sequence of layers, which makes it hard to use with models that reuse a layer (e.g., transformers).

**Sequential Models** Table 1 shows the results. Pfeife outperforms the state-of-the-art tools in all but one case. The odd lines in ViT have higher bubble/computation ratio, and therefore there are more opportunities for optimizing the schedules and thus the performance gap is wider.

Pfeife outperforms state-of-the-art tools mainly for two reasons: prefetching and looped schedules. Figure 7 shows the profiles of 4 GPUs training the ViT-g/14 model with Colossal-AI and Pfeife with and without loops and prefetch. When we use a naive 1F1B schedule (top two profiles), there is limited opportunity to exploit uneven slicing due to the fixed critical path. By prefetching some forward passes,

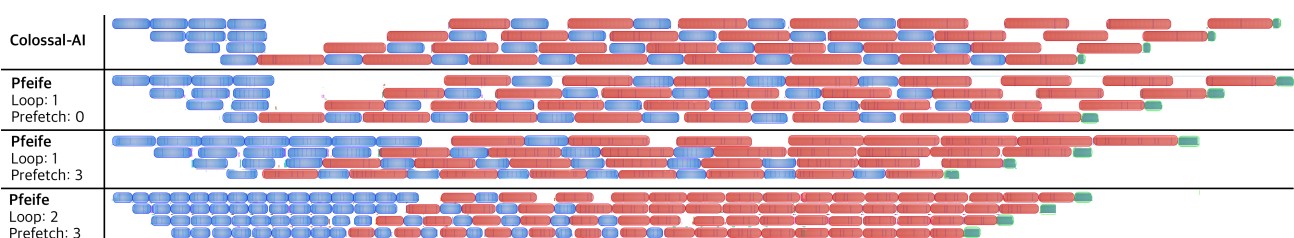

Figure 7: Profiling of ViT training with 4 GPUs and $B = 8$. Blue, red, and green blocks: forward, backward, and optimizer steps.

additional idle time (blank spaces) appears between the backward passes on the first device. Pfeife extends the forward pass on the first device to fill these idle slots, resulting in an 8% speedup. When combined with a looped schedule, the idle time at the beginning is further reduced (by 9%), bringing the total latency reduction to 16%.

Since Transformer-based models share embedding weights at both input and output layers, building a pipeline by hand with DeepSpeed is not straightforward. Our attempt to use ZeRO-DP resulted in out-of-memory errors, showing the limitations of data parallelism for models of this size. Pfeife, in contrast, successfully trained LLama and achieved up to 10% higher throughput than Colossal-AI. This demonstrates Pfeife's ability to handle large models more efficiently than existing parallelism approaches.

The only case in which Pfeife underperforms relative to another framework is with the ViT model using 8 devices and a batch size of 16, where DeepSpeed is faster. As the batch size increases and pipeline bubbles consequently decrease, performance increasingly reflects DeepSpeed's maturity and extensive engineering optimizations. However, the number of mini-batches per iteration (i.e., gradient accumulation steps) is inherently limited by model accuracy considerations and interactions with other parallelism strategies.

**Non-sequential models & mixing with Data Parallelism**
Pfeife can train complex models and be combined with other parallelism methods such as data parallelism. For example, SDXL uses a pre-trained autoencoder in front of the U-Net model. Due to its complex training pass and multiple skip-connections between its front and latter parts, U-Net models are not typically a target for pipeline parallelism.

We parallelized the trainable U-Net submodule in SDXL; Table 2 shows the results. We compared Pfeife with Deep-Speed's ZeRO-DP. The results show that Pfeife is comparable to ZeRO2. With more batches, Pfeife is sometimes faster than ZeRO2 since the proportion of the pipeline bubble is reduced. Also, Pfeife is 3-4 times faster than ZeRO3 and does not raise OOM in FP32.

These results demonstrate that Pfeife enables training of

significantly larger models that cannot be trained with other frameworks while maintaining comparable performance.

## 5. Conclusion

In this paper we presented Pfeife, the first tool that integrates transparently with PyTorch to provide fully automatic pipelining. It works by (1) capturing the whole data-flow graph of ML models through PyTorch's new JIT compiler, (2) distribute operations across devices and schedule the input data into mini-batches to establish the order in which they will travel the pipeline, and (3) intercept the execution of models and run their pipelined version instead.

We show that Pfeife runs large models that do not fit in a single GPU transparently, thus saving a significant amount of manual work. Our fine-grained, general pipeline optimizer enables Pfeife to train models up to 22% faster than state-of-the-art tools.

## Acknowledgements

This work was supported in part by national funds through FCT, Fundação para a Ciência e a Tecnologia, under project UIDB/50021/2020 (DOI: 10.54499/UIDB/50021/2020), and a National Research Foundation of Korea (NRF) grant funded by the Korean Ministry of Science and ICT (MSIT) (Grant No. RS-2024-00355459).

We acknowledge EuroHPC Joint Undertaking for awarding us access to Vega at IZUM, Slovenia.

## Impact Statement

This work aims to advance the field of machine learning, specifically in the area of large-scale model training. We are not aware of any specific societal or ethical concerns arising from this work, beyond the standard considerations that apply to the development and deployment of machine learning technologies.

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

# A. Implementation Details

## A.1. Implementation of Partial Pipeline Parallelism

```python
1  import torch
2  from some_library import PreProcessor, MainModel, get_dataset
3  from pfeife import initialize_pfeife, pfeife_compiler, PipeOptions, PipelineRunner
4
5  initialize_pfeife()
6
7  preproc_model = PreProcessor() # frozen preprocessing module
8  main_model = MainModel()
9
10 optimizer = torch.optim.Adam(main_model.parameters(), lr=1e-5)
11 criterion = torch.nn.CrossEntropyLoss()
12 dataset = get_dataset()
13
14 options = PipeOptions.from_args(cmd_args)
15 runner  = PipelineRunner(options, optimizer)
16 main_model = torch.compile(main_model, backend=pfeife_compiler)
17
18 for inputs, labels in dataset:
19     """ Original training loop:
20     with torch.no_grad():
21         inputs = preproc_model(inputs)
22     outputs = model(inputs)
23     loss = criterion(outputs, labels)
24     loss.backward()
25     optimizer.step()
26     print(f"loss: {loss.item()}")
27     """
28
29     # Do the same with a closure with inputs and labels
30     def iter_fn(inputs, labels):
31         with torch.no_grad():
32             inputs = preproc_model(inputs)
33
34         # The execution of the model is automatically pipelined
35         outputs = main_model(inputs)
36         loss = criterion(outputs, labels)
37         loss.backward()
38         print(f"loss: {loss.item()}") # allows side-effects
39         return loss # returned to runner.step()
40
41     # Set the target of pipelining
42     runner.set_exec_fn(iter_fn)
43
44     # Execute the pipeline by slicing `inputs` and `labels` into N micro-batches
45     losses = runner.step(inputs, labels)
46     loss = sum([loss.item() for loss in losses])
```

Listing 1: Example training loop using Pfeife.

Listing 1 shows an example of the full code required to train a model with Pfeife. It demonstrates Pfeife's strength: easy integration and the ability to mix multiple modules, some of which may be frozen. As we can see from the example, pipelining a training loop using Pfeife requires changing less than a dozen lines.

Before training, we compile the main module to be pipelined with Pfeife (line 16). The Pfeife runtime inserts a synchronization point in the forward function of the compiled module and attaches a backward hook to the result of the forward function. When input tensors or gradients flow to the pipelined module, Pfeife automatically sends the tensors to the pipelined devices and workers.

The core part of Pfeife involves enclosing forward to backward passes in a training loop within a closure (lines 29-39). We must enclose the training loop with a closure for two reasons: first, we want to execute the pre-processing part with a sliced

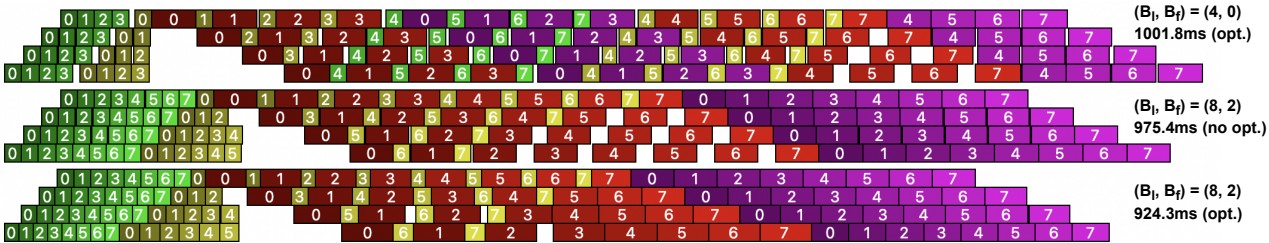

Figure 8: Three schedules of ViT-g/14 training with different $B_l$ and $B_f$. The second schedule uses uniform slicing in terms of the forward pass running time, while the other two are optimized by beam search.

micro-batch rather than a full mini-batch. Second, we want to control the timing of the execution of the pre-processor so that we can run the pre-processing part on the main device concurrently with the pipelined parts on the other devices.

## B. Scheduling Parameters

### B.1. Tradeoffs about Scheduling Parameters

Pfeife restricts the search space of our scheduling algorithm with a combination of scheduling parameters that simulates a looped-1F1B schedule. The tradeoff of each parameter is as follows:

- $(B)$ Total batch count: Enhances throughput per training iteration if increased, but can harm the performance and convergence if the batch size is too large.

- $(N_l)$ Loop count: Reduces the pipeline bubble at the beginning, but requires higher activation memory in the last device.

- $(B_l)$ Loop batch count: Communication latency can be hidden if increased, but uses significantly higher activation memory.

- $(\vec{B}_f)$ Prefetch batch count: Reduces the pipeline bubble between forward and backward passes, but requires higher activation memory in the first device.

For example, the schedule of Figure 3 suffers from high memory consumption due to storing many activations. The device at the bottom accumulates all the activations of 4 mini-batches in the first part of the loop. To mitigate this problem, we can reduce the number of mini-batches $B_l$ which are fed to a single loop. To fully utilize all the devices, $B_l$ should be a divisor of $B$ and $B_l \geq |\mathcal{D}|$.

We now emphasize the role of prefetching the forward passes before the first backward pass. Each device must run $(N_l - 1) \times B_l + i$ forward passes before executing the first backward pass where $i$ is the index of the device (indicated with the red vertical line). This configuration has the lowest peak memory usage for the looped 1F1B schedule.

$\vec{B}_f$ specifies that each device should execute a few more forward passes before the first backward pass. There are two positive effects of prefetching: (1) hide communication latency, and (2) fill in otherwise empty slots (the pipeline bubble). For example, observe the nodes $v_0^{+7}$ and $v_1^{+7}$ (the two light red "7" blocks of the two bottom devices). If we execute $v_0^{+7}$ before $v_0^{-1}$ (the bold green "1" block at the bottom), we can hide the latency between $v_0^{+7}$ and $v_1^{+7}$ which passes the result of the first device to the second device.

Figure 8 shows a concrete example with the actual timings of ViT-g/14. Through repeated local optimization with prefetching by beam search, we effectively fill the pipeline bubbles between forward and backward passes with a looped schedule. This level of optimization is achievable only with a fine-grained, operation-level pipeline. The subtle differences between forward passes in the second and third schedules demonstrate this precision.

### B.2. Examples of Pipeline Schedules

To show the expressiveness of this setting, we summarize the schedules proposed in previous work with parameters $(B, N_l, B_l, \vec{B}_f)$ and assuming 4 devices:

- GPipe (Huang et al., 2019), Figure 2: (8, 1, 8, (4, 5, 6, 7))
- 1F1B-PipeDream (Narayanan et al., 2019), Figure 8: (7, 1, 8, (0, 1, 1, 0))
- 1F1B-DAPPLE (Fan et al., 2021), Figure 3: (6, 1, 6, (0, 0, 0, 0))
- TeraPipe (Li et al., 2021), Figure 2: (8, 2, 4, (0, 1, 2, 3))
- Megatron-LM2 (Looped-1F1B) (Narayanan et al., 2021), Figure 4: (8, 2, 4, (3, 2, 1, 0))
- BFS Pipeline (Lamy-Poirier, 2023), Figure 4: (8, 4, 4, (4, 5, 6, 7))

We note that there are some scheduling strategies that cannot be expressed in this framework, namely kFkB (Wang et al., 2023c) and Chimera (Li & Hoefler, 2021). However, these strategies double the number of activations or weights, and thus they require a lot of memory, meaning they are not useful for the current memory-heavy models.

Nevertheless, our optimization algorithm is not specific to any pipelining strategy. These strategies can be expressed with our scheduling notation. If a user wants to consider those strategies, they just need to specify the strategy in our notation and give it to the optimizer.

## C. Coverage Test Details

Table 3: Program coverage of TorchBench.

| Total | Success | Multiple graphs | Model sharing error | Dynamo error | Incorrect result |
|-------|---------|-----------------|---------------------|--------------|------------------|
| 49    | 37      | 5               | 4                   | 2            | 1                |

Our coverage test compares the result (i.e., loss) between a pipelined model and the vanilla model. Also, we execute 5 training loops and compare their results after the parameters are trained.

Table 3 summarizes results of the coverage test. As we allow graph breaks in `torch.compile`, one of the model (`hf_Whisper`) is trained correctly even if it returns 5 subgraphs while its compilation.

However, 5 models result in tens of graphs so they raise an error from their side-effect handlers in the forward passes of the compiled module. Note that side-effect handlers connects the sub-graphs when there are graph breaks.

Four models raise an error while sharding sliced subgraphs to worker processes. Since we support multi-node training, we need to send a sliced PyTorch GraphModule through process pipes. However, their subgraphs cannot currently be marshaled by Python, preventing multi-process training.

Two models fail to compile with `torch.compile`. Since Pfeife depends on `torch.compile` to extract a static TorchFX graph, those models cannot be a target of pipeline parallelism until the PyTorch development team add supports for those models.

The last column refers to the `timm_efficientnet` benchmark. The cause of the error is that some parameters are not captured in the generated TorchFX GraphModule. This seems to be an engineering problem, and a fix is expected soon.

The complete list of the coverage test is as follows:

- Success: `alexnet`, `BERT_pytorch`, `dcgan`, `densenet121`, `functorch_dp_cifar10`, `hf_Albert`, `hf_Bert`, `hf_Bert_large`, `hf_clip`, `hf_DistilBert`, `hf_Roberta_base`, `hf_GPT2`, `hf_GPT2_large`, `hf_Whisper`, `LearningToPaint`, `lennard_jones`, `llama`, `mnasnet1_0`, `mobilenet_v2`, `mobilenet_v3_large`, `phlippe_densenet`, `phlippe_resnet`, `pytorch_unet`, `resnet152`, `resnet18`, `resnet50`, `resnext50_32x4d`, `shufflenet_v2_x1_0`, `squeezenet1_1`, `timm_nfnet`, `timm_regnet`, `timm_resnest`, `timm_vision_transformer`, `timm_vision_transformer_large`, `timm_vovnet`, `torch_multimodal_clip`, `vgg16`,
- Multiple graphs: `hf_BigBird`, `hf_Longformer`, `hf_Reformer`, `hf_T5_generate`, `opacus_cifar10`,
- Model sharing error: `hf_T5`, `hf_T5_base`, `hf_T5_large`, `dlrm`,
- Dynamo error: `cm3leon_generate`, `hf_Bart`
- Incorrect result: `timm_efficientnet`

## D. End-to-end Model Details

Table 4: Models used for evaluation. Train memory (GiB) = weights + gradients + optimizer state. Total = Estimated total memory usage when training with a single mini-batch.

| Model | Batch size | Weights | Activations | Train | Total |
|---|---|---|---|---|---|
| ViT-g/14 | 24 | 3.1 | 21.1 | 12.3 | 33.4 |
| LLama2-7B | 1 (1024 tokens) | 25.7 | 15.3 | 103 | 118 |
| StableDiffusion-XL (FP16) | 1 (1024px) | 5.1 | 8.4 | 41.1 | 49.5 |
| StableDiffusion-XL (FP32) | 1 (1024px) | 10.3 | 13.2 | 41.1 | 54.3 |

Table 4 shows the breakdown of memory usage in our end-to-end benchmarks. We reused the initial mini-batch through the whole training iterations to measure the performance of parallelization only.

ViT-g/14 (Zhai et al., 2022) is a vision model that we use as an example of a computation-intensive workload. Llama2 (Touvron et al., 2023), a large language model, represents a memory-intensive workload. Stable Diffusion XL (Podell et al., 2023) is in between the two, exhibiting both significant computational demands and a complex computation graph.

We exclude the GPT2-Large (Radford et al., 2019) model from our end-to-end training experiments due to its relatively small size, which results in suboptimal utilization of the A100 GPU.

## E. Estimated vs Actual Run Time and Memory Consumption

Table 5: Accuracy of the run time and memory consumption estimations with 8 mini-batches. The columns are: $|\mathcal{D}|$: number of devices (and mini-batches); $L_B$: size of a mini-batch; $N_l$, $B_l$, $B_f$: scheduling parameters; expected/actual peak memory of the first and last devices (GiB); expected/actual running time (ms). (*): Rejected by the scheduler.

| Model | $|\mathcal{D}|$ | $L_B$ | $N_l$ | $B_l$ | $B_f$ | Est. peak mem. | Act. peak mem. | Est. time | Act. time |
|---|---|---|---|---|---|---|---|---|---|
| ViT | 2 | 6 | 1 | - | 1 | 20.5 / 11.4 | 20.2 / 11.2 | 1899 | 1960 |
| ViT | 2 | 6 | 2 | 4 | 1 | 19.1 / 14.8 | 18.8 / 14.6 | 1808 | 1891 |
| ViT | 2 | 6 | 3 | 4 | 1 | 19.0 / 16.6 | 18.3 / 15.9 | 1776 | 1856 |
| ViT | 4 | 6 | 1 | - | 2 | 17.4 / 6.21 | 17.1 / 5.96 | 1121 | 1149 |
| ViT | 4 | 6 | 2 | 8 | 3 | 18.5 / 14.6 | 17.6 / 13.2 | 966 | 1002 |
| ViT | 4 | 6 | 3 | 8 | 3 | 20.6 / 18.7 | 19.2 / 16.8 | 936 | 974 |
| ViT | 8 | 6 | 1 | - | 3 | 11.6 / 3.33 | 10.8 / 3.05 | 760 | 768 |
| ViT | 8 | 6 | 2 | 8 | 3 | 13.5 / 8.27 | 11.7 / 7.37 | 621 | 683 |
| ViT | 8 | 6 | 3 | 8 | 3 | 13.8 / 10.3 | 12.0 / 9.10 | 591 | 697 (*) |
| GPT2 | 2 | 2 | 1 | - | 0 | 16.4 / 11.4 | 16.9 / 11.6 | 1230 | 1283 |
| GPT2 | 2 | 2 | 2 | 2 | 1 | 19.6 / 13.9 | 20.1 / 13.8 | 1142 | 1190 |
| GPT2 | 4 | 2 | 1 | - | 2 | 19.4 / 5.25 | 19.9 / 5.45 | 699 | 725 |
| GPT2 | 4 | 2 | 2 | 8 | 2 | 20.5 / 15.3 | 20.4 / 14.6 | 622 | 726 |
| GPT2 | 8 | 2 | 1 | - | 2 | 13.0 / 4.07 | 12.5 / 3.96 | 486 | 661 |
| GPT2 | 8 | 2 | 2 | 8 | 3 | 12.2 / 9.15 | 9.57 / 5.32 | 420 | 744 (*) |

Table 5 summarizes the results of the cost model accuracy for two models, ViT-g/14 and GPT2-large, with 512 tokens per batch. We run these models with a varying number of devices and batch sizes. For each setting, we let Pfeife find the best scheduling parameters, and record the actual peak memory and running time.

Regarding loop count ($N_l$), we can see in Table 5 that the performance increases with more loops, as it reduces the pipeline bubble. However, more loops also mean increased communication latency and also smaller kernels and thus higher overhead.

The estimated running time is very accurate, modulo a constant 30-60 ms warm-up time to distribute the mini-batches. However, when the slicing is too fine-grained (less than 10 ms forward time per node), each node finishes too quickly, and thus the Python threads cannot queue CUDA kernels fast enough. In this case, the overhead between the threads and CUDA dominates the running time, hence we made the optimizer reject schedules with an estimated running time of the forward pass less than 6 ms (marked with *).

