# OpenReview forum: "Pfeife: Automatic Pipeline Parallelism for PyTorch"
_ICML.cc/2025/Conference — ICML 2025 poster_

### Official Review · Reviewer_8hKh · 2025-03-11

**Overall Recommendation:** 2

**Summary:**

This paper proposes Pfeife, an automated tool that integrates with PyTorch to transparently partition and pipeline large machine learning models across multiple GPUs. It leverages PyTorch's JIT tracing to construct a data-flow graph of the model and then optimizes the pipeline schedule. Experimental evaluations show that Pfeife can outperform existing pipelining tools by up to 22% while handling complex, non-sequential models.

**Claims And Evidence:**

- Any specific reasons are provided for TorchTitan's limitation to LLMs only, yet the described techniques could be applied to arbitrary models. Please explicitly state what TorchTitan cannot do that Pfeife can.
- Distributed training is not solely due to limited memory capacity but also driven by computation requirements. Techniques like quantization and activation checkpointing reduce memory usage, so using memory constraints as the main motivation for pipeline parallelism is insufficient.
- Please specify what kind of data parallelism you are talking about. Traditional data parallelism requires minimal communication (just gradient averaging). When you refer to "it requires full synchronization of the weights across all devices", I suppose you are referring to ZeRO3/FSDP. Please cite them explicitly.
- The usage of "model parallelism", "tensor parallelism", and "pipeline parallelism" is inconsistent. For example, "For model and tensor parallelism" implies exclusivity, yet you also say "we focus on pipeline parallelism, which is a particular form of model parallelism." Please ensure consistent usage: when referring to model parallelism, it should include tensor parallelism and pipeline parallelism.
- Alpa does not require users to rewrite the model, and it is based on JAX instead of PyTorch.

**Essential References Not Discussed:**

Most cited works in the related section are at least two years old. Please incorporate more recent references, such as [B], [C], and [D], and clarify how Pfeife compares or improves upon these newer approaches. They all analyze or consider different kinds of parallelism in a distributed setting. Also, as mentioned in the above question, it would be good to discuss the applicability of Pfeife to DualPipe proposed in [A].
* [A] DeepSeek-AI, "DeepSeek-V3 Technical Report", https://arxiv.org/pdf/2412.19437v1
* [B] Chang Chen, Xiuhong Li, Qianchao Zhu, Jiangfei Duan, Peng Sun, Xingcheng Zhang, Chao Yang, "Centauri: Enabling Efficient Scheduling for Communication-Computation Overlap in Large Model Training via Communication Partitioning", ASPLOS, 2024.
* [C] Hongzheng Chen, Cody Hao Yu, Shuai Zheng, Zhen Zhang, Zhiru Zhang, Yida Wang, "Slapo: A Schedule Language for Progressive Optimization of Large Deep Learning Model Training", ASPLOS, 2024.
* [D] Ziheng Jiang et al., "MegaScale: Scaling Large Language Model Training to More Than 10,000 GPUs", NSDI, 2024.

**Experimental Designs Or Analyses:**

The experimental section is weak, as it includes only a small set of models and configurations.
- Pipeline parallelism is typically used in multi-node environments, where tensor parallelism may be used within a node, and pipeline parallelism spans multiple nodes. How is communication modeled in a multi-node environment? Do you have any results on the multi-node setting?
- The comparison with automatic parallel frameworks like DeepSpeed and Colossal-AI is insufficient. PiPPy also has an automatic mode for pipeline parallelism (https://pytorch.org/docs/main/distributed.pipelining.html#option-2-splitting-a-model-automatically) —please compare with that as well. Additionally, comparisons to manually optimized systems like TorchTitan and Megatron-LM would show how Pfeife fares against specialized manual implementations.
- 3D parallelism is prevalent in nowadays large model training. The experiments only contain DP+PP results. What about including TP? Can your PP method incorporate tensor parallelism?
- Only one specific configuration (ViT, 4, 4) shows significant improvements (over 20%). Why is this configuration particularly favorable? Why should Pfeife generalize well to other scenarios?
- What is the cost of profiling? For example, if users want to run a job on an 8-node cluster, do they need to firstly launch the profiling job for this 8-node cluster? This would be very time-consuming and costly.
- See the below question for C.1, how do you handle faster, newer-generation GPUs? If latency estimations are prone to fail under extreme speed, does that mean Pfeife cannot be generalized to other devices?

**Methods And Evaluation Criteria:**

Only three models (ViT, LLaMA, and Stable Diffusion) are evaluated. Beyond Stable Diffusion, can Pfeife handle other model types, such as MoE models (e.g., DeepSeek-V3) or state-space models (e.g., Mamba)?

**Other Comments Or Suggestions:**

See the above sections.

**Other Strengths And Weaknesses:**

See the above sections.

**Questions For Authors:**

See the above sections.

**Relation To Broader Scientific Literature:**

This paper mainly focuses on automating the pipeline parallelism process in distributed training, which can potentially reduce programming efforts for deploying models in a distributed environment.

**Theoretical Claims:**

- How do you decide which parts of a model can be parallelized? For instance, in a Stable Diffusion pipeline, do users have to manually define which parts go into different pipeline stages, as shown in Listing 1?
- Many HuggingFace models cannot be fully traced by TorchDynamo. How do you handle multiple partial graphs? Do you use a custom torch.fx tracer to deal with such models?

---

> ### Author Rebuttal · Authors · 2025-03-26
>
> We would to thank the reviewers' time and feedback.
>
>
> Reviewer aSA1
>  - Unfortunately, we only have access to the 2 machines we used in the experiments.
> We don't have budget for more.
>
>  - Regarding correctness, first we note that we run an order of magnitude more
> models than most academic papers. It required substantial engineering effort to
> ensure the results were correct and that we supported most features of PyTorch's
> torch.compile/TorchFX. These are still under development and thus a moving target.
> As such, we believe we deserve some credit for being able to run so many models
> and being open that not all models work (we could have simply left these out).
> As shown in table 3, only 1 out of 49 models produces incorrect results,
> which is actually due to a limitation in the model capturing of torch.compile.
> It's not a bug in our algorithm or implementation.
>
>  - "Also, the A100 cluster is too powerful to benchmark multiple sets of pipeline
> parameters (i.e., forward time of each computation node is too short to correctly
> estimate total latency as we increase loop count or number of devices). Therefore,
> we need to limit the performance of a device itself."
>
> Apologies for the broken English; we'll fix that.
> What we mean is that some configuration parameters generate forward passes that
> are too small. At some point, the overhead of spinning a new GPU kernel and
> the communication starts to dominate, and then the results are not interesting
> anymore. As we mention in the paper, we prevent the scheduler from generating
> these configurations since they are unlikely to yield good performance.
> The slower the device the more interesting configurations we can test.
> Anyway, this is nothing fundamental, we just wanted to try all configurations
> we could run.
>
>
> Reviewer wSiC
>  - We run an order of magnitude more models than most academic
> papers. We believe we deserve some credits for that.
> We have also run a comprehensive study of the optimal schedules to show
> that unbalanced schedules are better. We understand we didn't run experiments
> with a thousands GPUs due to not having them, but our experiments meet or
> even exceed the bar set by previous pipelining papers. And we do offer new
> insights into the problem and a new algorithm.
>
>  - We compared with the systems that are available. GPipe and PipeDream are
> not. FSDP requires manual work and it overlaps significantly with
> DeepSpeed. We believe the experiments we did are sound and provide good insights.
>
>
> Reviewer NzCV
>  - Profiler: this is done only once when the model starts and is done per
> operation. Hence, the overhead is negligible for long-running training
> sessions. The time it takes is about the same as running the model
> sequentially.
> Profiling of inter-device communication is done only once on installation.
>
>  - Limitations: Pfeife's current implementation only supports natively
> looped schedules. The algorithm itself is obliviously of the schedule,
> but we use a template to produce looped schedules. We would like to
> have a language to specify templates for schedules. That would be a great
> extension to this work.
>
>  - Using Pfeife from other frameworks: Pfeife has a frontend part that
> reads TorchFX graphs. That part is specific to PyTorch, but it's thin;
> we only need to create a graph with nodes.
> Then, we have the execution engine that has some bits tied to PyTorch
> since we implement a backend for torch.compile. But it can as well be
> replaced.
> The core, the scheduling algorithm, etc, they are framework independent
> since they operate over model graphs annotated with costs.
>
>
> Reviewer 8hKh
>  - We don't have results for multi-server deployments. We only have 2
> servers that are shared.
>
>  - PiPPy is a work in progress. It is not ready for comparisons (we tried).
>
>  - We could compare with TP, but this requires more manual work than we
> could afford.
>
>  - Pfeife's scheduling algorithm can handle DualPipe. Our current implementation
> is hardcoded with a scheduling template that spawns looped schedules only.
> But that's not a fundamental limitation, we just didn't implement other
> templates. The scheduling algorithm itself is oblivious since it operates over
> graphs. Implementing other templates can be done with a dozen lines of code.
>
>  - Pfeife can work with powerful GPUs. We are sorry for the broken English.
> What we meant is that we should not slice graphs too thinly for faster devices
> since then the overhead of launching GPU kernels and communication starts to
> dominate.
>
>  - We don't include results for other models mostly because we didn't have
> enough compute credits. Running a couple of times to test correctness is one
> thing, but taking stable performance results for multiple configurations
> and attempting to compare against other frameworks requires a lot more
> compute. Another reason is that the other frameworks require manual work
> for each model, and our team is small. We cannot spend months trying to
> run a dozen models in multiple frameworks. Plus most tools are fragile
> and crash often.

---

### Official Review · Reviewer_NzCV · 2025-03-13

**Overall Recommendation:** 2

**Summary:**

The paper introduces Pfeife, a new tool that integrates with PyTorch to provide automatic pipelining of machine learning models.
Pfeife aims to address the memory limitations of GPUs when training large models by parallelizing the execution of these models across multiple devices.
It leverages PyTorch's tracing JIT compiler (TorchDynamo) to capture a static data-flow graph of a PyTorch model and then automatically generates a pipeline schedule to distribute the operations across devices.
The authors claim that Pfeife can run large models that would otherwise not fit on a single device and that it outperforms state-of-the-art pipelining tools.

**Claims And Evidence:**

Claim 1. Pfeife can run large models that would otherwise not run due to not fitting in a single device.
The paper provides empirical results demonstrating Pfeife's ability to run large models. This advantage is shared by all parallelism techniques, which is not an exclusive benefit of Pfeife.

Claim 2. Pfeife outperforms state-of-the-art tools by up to 22% in terms of training throughput.
The paper presents comparative evaluations against other pipeline tools. This number is from Table 1.

Claim 3. Pfeife can pipeline non-sequential models such as Stable Diffusion that are not supported by previous pipeline parallelism tools. The paper mentions this capability, and the evaluation includes Stable Diffusion. The authors should provide a more detailed explanation of why previous tools cannot support such models and how Pfeife overcomes these limitations.

**Essential References Not Discussed:**

n/a

**Experimental Designs Or Analyses:**

The experimental design involves evaluating Pfeife on large ML models and comparing its performance with existing pipelining tools.

The paper mentions using a profiler to measure computation time and memory consumption.  More details on the profiling methodology, including the granularity of the profiling (e.g., per-operation), the profiling overhead, and the accuracy of the profiler, would strengthen the analysis.

The authors acknowledge that linearizing the computation graph is an approximation.  A more detailed analysis of the impact of this approximation on the generated schedules would be valuable.

**Methods And Evaluation Criteria:**

Pfeife leverages PyTorch 2's TorchDynamo to capture the data-flow graph of ML models. It employs a pipeline scheduling algorithm that considers both graph slicing and scheduling parameters to optimize performance.
The authors use a cost model based on critical path analysis to estimate running time and memory usage.

The evaluation includes experiments with large models, and the primary performance metric appears to be running time or throughput.  It would be beneficial to include memory usage as a key evaluation criterion, given that the motivation is to address memory limitations.  The choice of benchmark models seems appropriate for demonstrating the capabilities of the system.

**Other Comments Or Suggestions:**

n/a

**Other Strengths And Weaknesses:**

Strengths:
* The paper addresses an important problem in deep learning: memory limitations in training large models.
* Pfeife offers an automated and transparent way to pipeline models, which can improve usability.
* The approach of co-optimizing slicing and scheduling is novel and promising.

Weaknesses:
* The theoretical claims and the cost model would benefit from more rigorous analysis.
* The limitations of the approach, such as those arising from the linearization of the computation graph, should be discussed in more detail.

**Questions For Authors:**

1. What are the limitations of Pfeife, and how do you plan to address them?
2. How difficult can we apply Pfeife to other machine learning frameworks and compilers, such as JAX/XLA?

**Relation To Broader Scientific Literature:**

The paper effectively situates Pfeife within the context of existing parallelization techniques for machine learning models, including data parallelism, weight sharding, model parallelism, and tensor parallelism. It provides a good overview of pipeline parallelism and discusses relevant prior work such as PipeDream, GPipe, DAPPLE, AutoPipe, TeraPipe, and others. The authors clearly articulate how Pfeife builds upon and differs from previous approaches, such as by using a more fine-grained partitioning (per operation) and co-optimizing slicing and scheduling.

**Theoretical Claims:**

The paper describes synchronous and asynchronous pipeline parallelism and an algorithm to find pipeline schedules.  The authors also investigate optimal workload distribution conditions.

The paper uses a cost model based on critical path analysis.  This cost model and the scheduling algorithm are central to the paper's contributions. The correctness of the algorithm and the accuracy of the cost model in predicting performance should be rigorously established, possibly with proofs or theoretical analysis in the supplementary material.

I suggest to avoid using the term "optimal" unless there are solid proof for that claim.

---

### Official Review · Reviewer_wSiC · 2025-03-13

**Overall Recommendation:** 2

**Summary:**

The paper introduces Pfeife, a system integrating with PyTorch's `torch.compile` to automate pipeline parallelism without user intervention. Pfeife partitions models at an operation-level granularity across multiple GPUs, employing a cost model combined with beam search to optimize pipeline scheduling. Key claimed contributions include a novel algorithmic framework allowing simultaneous optimization of pipeline slicing and scheduling, demonstrating that uneven workload distribution with prefetching can improve performance significantly. Evaluations demonstrate Pfeife achieving throughput improvements of up to 22% over DeepSpeed and Colossal-AI across various large models, including ViT-g/14, Llama2-7B, and StableDiffusion-XL.

**Claims And Evidence:**

Claims about performance improvement and automation are generally supported through rigorous experiments comparing Pfeife to DeepSpeed and Colossal-AI.

**Essential References Not Discussed:**

Not necessarily essential, but GraphPipe is of interest to this work:

Jeon, Byungsoo, et al. "Graphpipe: Improving performance and scalability of dnn training with graph pipeline parallelism." arXiv preprint arXiv:2406.17145 (2024).

**Experimental Designs Or Analyses:**

Experimental setups using ViT-g/14, Llama2-7B, and StableDiffusion-XL are adequate for evaluating memory management and throughput in large-scale models: they cover a wide range of recent architectures and are of adequate size.

**Methods And Evaluation Criteria:**

The evaluation methods are well designed and carried out for demonstrating Pfeife’s practicality and efficiency within PyTorch. However, the limited scope in evaluation criteria—lacking comprehensive comparisons with established frameworks like GPipe, PyTorch FSDP and PipeDream—undercuts the robustness of its methodological validation. Moreover, Pfeife’s reliance on PyTorch's compiler limits the scope of its application, especially considering the reported failure rates on several TorchBench models.

**Other Comments Or Suggestions:**

The author may consider making the figures more b/w friendly.

**Other Strengths And Weaknesses:**

Strengths:

Pfeife's integration with PyTorch’s TorchDynamo presents practical usability.

The idea of automatic processing in lieu of manual annotation is fresh and interesting.

Weaknesses:

Pfeife lacks fundamental novelty - there is a plethora of existing pipeline parallel systems, and in the absence of strong, comprehensive benchmarking results backed by available artifacts, it is not a strong submission to this venue.

**Questions For Authors:**

I do wonder if the author has a reason for omitting other well-known pipeline parallel systems like GPipe and PipeDream in the work. Since the manuscript is built on PyTorch, a comparison with FSDP may also be warranted. There is also more recent works on more sophisticated flavors of pipeline parallelism, such as GraphPipe.

**Relation To Broader Scientific Literature:**

Pfeife situates itself within the context of automatic pipeline parallelism, effectively highlighting distinctions from manual annotation-required systems like PiPPy and Varuna. However, the literature review fails to deeply engage with critical predecessors such as GPipe, PipeDream, GraphPipe, and ZeRO-3. Particularly, the omission of GraphPipe (which introduced DAG-based scheduling with significant performance gains) severely undermines the perceived novelty of Pfeife's algorithmic contributions.

**Theoretical Claims:**

The manuscript does not contain substantial new theoretical results but correctly refers to established concepts such as cost models of pipeline parallelism. No formal proofs requiring validation are presented; theoretical claims referenced from literature are appropriate and correctly applied.

---

### Official Review · Reviewer_aSA1 · 2025-03-13

**Overall Recommendation:** 3

**Summary:**

This paper presents Pfeife, a tool that automatically performs pipeline parallelization of PyTorch models. Compared to prior methods, the main innovation is that the pipelining is performed in a manner that is completely transparent to the developer, requiring no manual annotations. Specifically, Pfeife is implemented as a backend to $torch.compile$. Starting from a computation graph traced by the PyTorch compiler, Pfeife uses an estimated cost model tuned to the specific hardware, and performs a co-optimizing search that alternates between fusion ("slicing") and pipeline scheduling, aiming to minimize the critical path. Experiments with an 8xA100 (40GB) cluster shows that Pfeife exhibit improvements over DeepSpeed and ColossalAI tools on Llama2-7b and is comparable on SDXL, despite requiring no manual effort.

**Claims And Evidence:**

This paper makes two main claims
1) Pfeife performs pipelining with no manual effort. This is generally true: the schedules are generated without any manual efforts, and the schedules are generally (though not always) correct from the coverage tests.
2) Pfeife discovers pipeline schedules that are as good as semi-automated pipeline tools. This is also supported by the experimental results, albeit for relatively "small" models (Llama2-7b and SDXL)

**Essential References Not Discussed:**

N/A

**Experimental Designs Or Analyses:**

The way the experiments are properly designed (both the correctness and performance) and the analysis all make sense.

However, I would have liked to see at least one hand-tuned pipeline as a baseline for comparison.

**Methods And Evaluation Criteria:**

The proposed methods and evaluation both make sense.

It would have been nice to evaluate performance on even larger models / diverse hardware topologies (where pipelining is even more necessary and tricky to get right), but I recognize that this is out of the reach for the majority of researchers.

**Other Comments Or Suggestions:**

For Table 4 in Appendix C.3, it would be good to have additional columns that display the errors.

**Other Strengths And Weaknesses:**

I think the contribution of completely automatic pipelining is strong. I can see the ideas here being incorporated into future works. It's also interesting that this work allows SDXL to be pipelined, though I'm unclear about the practical implications

The main weaknesses are
1) the co-optimization algorithm is quite complicated and the design decisions receive no justification, either theoretical or practical
2) the experiments are conducted in a relatively limited setting (8x A100s, llama 7b)
3) the correctness is somewhat low. Even being generous in what is counted as a bona fide failure, approximately 10% of the evaluation benchmark set (which consists of relatively small models) yields failures

**Questions For Authors:**

"Also, the A100 cluster is too powerful to benchmark multiple sets of pipeline parameters (i.e., forward time of each
computation node is too short to correctly estimate total latency as we increase loop count or number of devices). Therefore,
we need to limit the performance of a device itself." Can you clarify this statement?

**Relation To Broader Scientific Literature:**

This paper focuses specifically on pipeline parallelism for multi-GPU workloads. Compared to existing pipeline tools, the main novelty is that the pipelining is done in a completely automatic fashion, without any developer input. The cost model is inspired by DAPPLE (Fan et al., 2021). The search is unique in that, for every pipeline schedule in the scheduling search space, a beam search is conducted for an optimal graph slice; the search space subsumes many (but not all) prior pipeline schedules (1F1B).

**Theoretical Claims:**

N/A

---

### Decision · Program_Chairs · 2025-05-01

**Decision:**

Accept (poster)

**Comment:**

Reviewers consistently point to the utility of an automated pipeliner.  The rebuttal responds well to the criticisms: as the reviewers note, the experiments are on sufficiently large (and sufficiently many) models to give confidence that the technique works.  As the authors note, they are to be commended for listing the models on which it does not work.

The reviewers' numerical scores are not consistent with the review text, so the review text takes priority in decision making.